# Meropenem MICs at Standard and High Inocula and Mutant Prevention Concentration Inter-Relations: Comparative Study with Non-Carbapenemase-Producing and OXA-48-, KPC- and NDM-Producing *Klebsiella pneumoniae*

**DOI:** 10.3390/antibiotics12050872

**Published:** 2023-05-08

**Authors:** Maria V. Golikova, Elena N. Strukova, Kamilla N. Alieva, Vladimir A. Ageevets, Alisa A. Avdeeva, Ofeliia S. Sulian, Stephen H. Zinner

**Affiliations:** 1Department of Pharmacokinetics & Pharmacodynamics, Gause Institute of New Antibiotics, 11 Bolshaya Pirogovskaya Street, 119021 Moscow, Russia; kindyn@yandex.ru (E.N.S.); qvimqwem@yandex.ru (K.N.A.); 2Pediatric Research and Clinical Center for Infectious Diseases, 9 Prof. Popov Street, 197022 St. Petersburg, Russia; ageevets@list.ru (V.A.A.); avdeenko-alya@mail.ru (A.A.A.); sulyan1994@mail.ru (O.S.S.); 3Department of Medicine, Harvard Medical School, Mount Auburn Hospital, 330 Mount Auburn St., Cambridge, MA 02138, USA; szinner@mah.harvard.edu

**Keywords:** antibiotic resistance, carbapenems, OXA-48 carbapenemase, NDM carbapenemase, KPC carbapenemase, meropenem, mutant prevention concentration, inoculum effect, *Klebsiella pneumoniae*

## Abstract

The minimal inhibitory concentration (MIC) is conventionally used to define in vitro levels of susceptibility or resistance of a specific bacterial strain to an antibiotic and to predict its clinical efficacy. Along with MIC, other measures of bacteria resistance exist: the MIC determined at high bacterial inocula (MIC_HI_) that allow the estimation of the occurrence of inoculum effect (IE) and the mutant prevention concentration, MPC. Together, MIC, MIC_HI_ and MPC represent the bacterial “resistance profile”. In this paper, we provide a comprehensive analysis of such profiles of *K. pneumoniae* strains that differ by meropenem susceptibility, ability to produce carbapenemases and specific carbapenemase types. In addition, we have analyzed inter-relations between the MIC, MIC_HI_ and MPC for each tested *K. pneumoniae* strain. Low IE probability was detected with carbapenemase-non-producing *K. pneumoniae,* and high IE probability was detected with those that were carbapenemase-producing. MICs did not correlate with the MPCs; significant correlation was observed between the MIC_HI_s and the MPCs, indicating that these bacteria/antibiotic characteristics display similar resistance properties of a given bacterial strain. To determine the possible resistance-related risk due to a given *K. pneumoniae* strain, we propose determining the MIC_HI_. This can more or less predict the MPC value of the particular strain.

## 1. Introduction

The minimal inhibitory concentration (MIC) is conventionally used to define in vitro levels of the susceptibility or resistance of a specific bacterial strain to an antibiotic and to predict its clinical efficacy [1]. However, even if a microorganism is initially susceptible to the antibiotic, the emergence of resistance during antibacterial therapy can lead to treatment failure. Resistance development during antibiotic exposure can be explained by the mutant selection window (MSW) hypothesis [2,3]. Accordingly, the selection of resistant cells occurs if antibiotic concentration falls into the MSW—the concentration range between the MIC (the lower MSW border) and the mutant prevention concentration, MPC (the upper MSW border). The applicability of the MSW hypothesis has been confirmed for several classes of antibiotics, including fluoroquinolones [4,5], glycopeptides [6,7], oxazolidinones [8,9], cephalosporins [10] and carbapenems [11,12], among others. The upper border of the MSW, the MPC, can predict the emergence of resistance of a given bacterial isolate during antibiotic exposure. Along with the MPC, the risk of antibiotic failure can be predicted by MICs determined at high microbial burdens of 10^7^ CFU/mL (HI, MIC_HI_), i.e., significantly higher than the 10^5^ CFU/mL (SI) used in standard in vitro susceptibility testing (MIC). This is recognized as the inoculum effect (IE) [13] that has been observed with several classes of antibiotics [14,15], especially with beta-lactams and beta-lactam/beta-lactamase inhibitor combinations, whose activity may be significantly decreased at high inocula [16,17]. With Gram-negative bacteria, the main drive of the IE with beta-lactams is the production of bacterial beta-lactamases that confer resistance at high bacterial concentrations. This has been well described in numerous in vitro studies with beta-lactamase-producing *Escherichia coli*, *K. pneumoniae*, *Pseudomonas aeruginosa* and *Acinetobacter baumannii* exposed to beta-lactams, including penicillins and aminopenicillins, cephalosporins, carbapenems and monobactams [16,17,18]. Published in vivo data using infected animals have shown decreased effectiveness of cefazolin, cefotaxime, piperacillin-tazobactam, amoxicillin-clavulanate and aztreonam due to the IE [19,20,21,22,23,24]. The clinical relevance of the IE has been confirmed in several studies of patients with staphylococcal bacteremia undergoing cefazolin therapy [25,26]. Understanding the role of the IE with beta-lactams is important in the treatment of high bacterial-burden infections, since decreased antibacterial activity may result in unexpected treatment failures.

In summary, MIC, MIC_HI_ and MPC represent the bacterial “resistance profile” that captures in vitro parameters of bacterial resistance responsible for treatment failures. The specific mechanism of resistance might be less important than the inoculum effect or MPC value with some antibiotics. To verify this assumption, we used *K. pneumoniae* strains that either produce or do not produce carbapenemases. It is important to determine pharmacodynamic characteristics for individual *K. pneumoniae* strains and to identify possible inter-relations among these susceptibility/resistance measures. Moreover, the ability to produce carbapenemases may influence MPC and MIC_HI_ values per se, and may predict resistance risks during meropenem therapy.

Meropenem is a commonly used beta-lactam in clinical practice to treat infections caused by Gram-negative bacteria, including those that produce beta-lactamases. *K. pneumoniae* produce a wide range of beta-lactamases, including the widespread OXA-48 [27], NDM (New Delhi Metallo-beta-Lactamase) [28,29] and KPC (*Klebsiella pneumoniae* carbapenemase) [30] carbapenemases. The prevalence of carbapenem-resistant *K. pneumoniae* clinical isolates in hospitals was repeatedly reported [31,32].

To determine the meropenem “resistance profiles” of *K. pneumoniae,* to explore the influence of carbapenemase type on MPC and MIC_HI_ values, and to find inter-relations among MIC, MIC_HI_ and MPC, *K. pneumoniae* clinical isolates and ATCC strains producing carbapenemases of different types (OXA-48, NDM and KPC), along with carbapenemase-negative strains (control group), were used in the study. In addition, we aimed to explore if the IE occurs with meropenem and how it depends on the bacterial production of different types of carbapenemases.

## 2. Results

### 2.1. Susceptibility Testing with Meropenem and K. pneumoniae at SI and HI

The results of susceptibility testing with meropenem and *K. pneumoniae* isolates at standard and high inocula were divided into four groups, depending on their ability to produce carbapenemases, and their specific types are presented in Table 1. Among each strain group (carbapenemase-negative, OXA-48, KPC and NDM carbapenemase producers), meropenem MICs varied widely from low (≤8 mg/L) to medium or high (up to 512 mg/L). At high inocula, most of the tested strains exhibited higher MIC values than at standard inocula; however, this MIC increase differed according to the production of specific carbapenemases. The most obvious inoculum-related meropenem MIC shift (from 8- to 256-fold) was exhibited by NDM-producing isolates and it was always associated with the IE. Compared to NDM producers, KPC-producing *K. pneumoniae* were characterized by a less pronounced MIC increase at high bacterial inocula (4- to 16-fold), and the IE was detected in four of the nine strains with MICs ranging from 8 to 32 mg/L; the most resistant strains with MICs of 64 and 128 mg/L did not exhibit an IE.

Among OXA-48-producing *K. pneumoniae* (8- to 64-fold MIC increases), IE occurred only in meropenem-susceptible isolates with MICs up to 8 mg/L; minimal 2-fold inoculum-induced decrease in meropenem activity occurred with the most resistant *K. pneumoniae* isolate (MIC of 32 mg/L). Non-carbapenemase-producing *K. pneumoniae* were not usually prone to the IE, except the two most meropenem-susceptible isolates with MICs of 0.03 and 0.06 mg/L; these strains showed 64- and 32-fold inoculum-related MIC decreases, respectively.

To explore if MICs at high and standard inocula are inter-dependent, the respective data, MIC versus MIC_HI_, for four *K. pneumoniae* isolate groups were plotted on a single graph (Figure 1).

The poor sigmoid relationship between the MIC and MIC_HI_ was observed only with carbapenemase-non-producing *K. pneumoniae* with *r*^2^ = 0.65. However, the low correlation coefficient indicates that non-carbapenemase-producing isolates did not show significant correlations between meropenem MICs at either inoculum densities. Carbapenemase-producing isolates did not show any correlations between meropenem MICs at the two inoculum densities. Therefore, it can be hypothesized that bacterial susceptibilities at different inocula are most likely not inter-related for carbapenemase-producing *K. pneumoniae* or that this relation is poor for carbapenemase-negative strains. However, data points were stratified depending on the *K. pneumoniae* group: lower MIC_HI_s were detected with carbapenemase-negative *K. pneumoniae*, relatively high MIC_HI_s are seen with OXA-48 carbapenemase producers and the highest MIC_HI_ values were detected with KPC and NDM producers. In general, meropenem activity against *K. pneumoniae* at high inocula depends on the ability of bacteria to produce specific types of carbapenemases.

Along with using absolute MICs at high and standard inocula for inter-MIC correlations, we analyzed the relationship between the initial MICs at SI and the ratio of MIC_HI_ to MIC (MIC shift, IE measure). This analysis was aimed at exploring whether IE occurrence depends on the isolate’s initial MIC and if it can be predicted based on MIC data alone (Figure 2). As seen in Figure 2, sigmoid “MIC_HI_/MIC ratio—MIC” relationships with high correlation coefficients (*r*^2^ = 0.86–0.92) were obtained for all isolate groups: the higher the isolate’s MIC at standard inocula, the lesser the inoculum-related MIC decrease and IE probability. As mentioned above, carbapenemase-non-producing *K. pneumoniae* were less prone to the IE than carbapenemase producers and were characterized by less inoculum-related MIC decreases. OXA-48 producers were not prone to the IE if the isolate MIC was higher than the meropenem MIC breakpoint; even with high meropenem MICs (up to 32 and 64 mg/L, respectively), KPC and NDM producers exhibited the IE at high inocula.

### 2.2. MPC Determination of Meropenem against K. pneumoniae

The MPC data for *K. pneumoniae* strains are presented in Table 1. MPCs varied over a wide range depending on the isolate and its ability to produce carbapenemase. The smallest MPC values were determined for carbapenemase-non-producing *K. pneumoniae*; with most strains the MPC ranged from 1 to 8 mg/L. These MPC values were very close to MIC_HI_ values. MPCs of carbapenemase-producing *K. pneumoniae* were significantly higher than carbapenemase-negative strains. However, considerable MPC differences among the strain groups were observed. For example, *K. pneumoniae* strains with equal meropenem MICs (8 mg/L) but producing different carbapenemase types (3111, 1904 and 2131), had MPCs that differed by an order of magnitude: 128 mg/L for OXA-48 producer versus 2048 for both KPC and NDM producers, respectively. In total, meropenem MPC range for OXA-48 producers was from 32 to 256 mg/L, while for KPC and NDM producers, it was from 1024 to 8192 mg/L.

To examine correlations between the MICs and the MPCs, the respective data for all *K. pneumoniae* strains were plotted on one graph (Figure 3). As seen in the figure, no correlation could be found between the MICs and the MPCs regardless of the isolate and its ability to produce any carbapenemases. However, there is some MPC data stratification among the strains: the lowest values were in carbapenemase-negative *K. pneumoniae*, were a little higher with OXA-48 producers and were higher still in KPC and NDM producers.

Assuming that both MIC_HI_ and MPC are the resistance measures that display “hidden” bacterial resistance potential, we plotted these strain characteristics on the same graph in an attempt to find any correlation between these resistance measures (Figure 4).

As seen in the figure, a distinct sigmoid relationship was observed between the MIC_HI_s and the MPCs based on the merged data for all *K. pneumoniae* strains with r^2^ = 0.93. Interestingly, the data points that belong to different *K. pneumoniae* groups (carbapenemase-negative, OXA-48, KPC and NDM producers) follow each other with respect to MIC_HI_ increases. The lowest MIC_HI_ and MPC values were seen with carbapenemase-non-producing *K. pneumoniae*; then, OXA-48 producers follow and the highest MIC_HI_ and MPC values were seen with KPC- and NDM-producing strains.

## 3. Discussion

In the current study, we have provided a comprehensive analysis of “resistance profiles” of *K. pneumoniae* strains that differ by meropenem susceptibility, ability to produce carbapenemases and specific carbapenemase types.

We found that meropenem susceptibility of carbapenemase-producing *K. pneumoniae* was significantly influenced by the bacterial inoculum density, and IE was most often detected with these isolates. In contrast, non-carbapenemase-producing *K. pneumoniae* isolates were not usually prone to the IE and inoculum-induced meropenem MIC increase was minimal except for in the two most susceptible strains. These results are concordant with many other in vitro studies where the beta-lactam IE was confirmed most often with beta-lactamase-producing Gram-negative bacteria [17]. In the current study, among carbapenemase-producing isolates, NDM producers were characterized by significantly higher inoculum-related MIC increases (up to the 256-fold); KPC- and OXA-48-producing strains exhibited relatively lower MIC shifts (Table 1).

Assuming the possible threat of IE when treating patients with infections caused by Gram-negatives, it would be valuable if this phenomenon could be predicted with given bacterial isolates and antibiotics in order to optimize therapy. The current study did not find a relationship between MIC and MIC_H_ for carbapenemase-producing *K. pneumoniae* (Figure 1). With the control group of non-carbapenemase-producing strains, a poor “MIC_HI_-MIC” correlation was observed (*r*^2^ = 0.65). Hence, MIC itself cannot predict specific MIC_HI_ values for a given *K. pneumoniae* strain. However, it was shown that the MIC shift at high inocula, i.e., MIC_HI_/MIC ratio, does correlate with the initial MIC (Figure 2); therefore, assuming the initial meropenem MIC value, we can predict possible decreases in meropenem susceptibility at high inocula. Interestingly, the higher the MIC of isolate at SI, the weaker the MIC decrease at high inocula was observed; this was seen with both carbapenemase-producing and non-carbapenemase-producing *K. pneumoniae.* From our previous results with imipenem and doripenem alone or in combination with the carbapenemase inhibitor relebactam, the IE was more likely when *K. pneumoniae* MIC was lower [33]. With highly carbapenem resistant *K. pneumoniae* isolates, the IE was not observed. This can be explained by potentiation of carbapenemase enzymes with inoculum increases in strains with lower initial MICs than in highly resistant strains which already possess high carbapenemase activity.

Unfortunately, IE was detected in all meropenem-susceptible carbapenemase producers. This causes concern about decreased meropenem clinical results in treating high-density infections caused by these organisms given that their meropenem MIC_HI_s exceed peak therapeutic plasma meropenem concentration with a standard dosing regimen (approximately 30 mg/L after a standard dose of 1 g intravenously every 8 h [34,35]) or exceed the meropenem concentrations for most of the dosing interval with a high-dose meropenem regimen (approximately 40 mg/L after a dose of 2 g intravenously every 8 h [36]).

The MPC is a well-known measure of bacterial ability to produce antibiotic-resistant cells. Based on the MSW hypothesis, the MPC value can be used to assess whether certain microbes pose a high risk of decreased antibacterial effectiveness due to the development of resistance. In the MPC-analysis of our *K. pneumoniae* strains, we detected mostly low MPC values for carbapenemase-non-producing strains (1–8 mg/L), while carbapenemase producers in general exhibited high MPCs (32 mg/L at minimum); therefore, the “hidden” resistance potential of such strains is very high. Among carbapenemase producers, the MPCs of OXA-48-producing isolates were prominently lower than those of KPC- and NDM-producing isolates (about an order of magnitude). Possibly, it can be explained by the ability of different bacterial cells, depending on carbapenemase type, to enhance carbapenemase gene expression under the influence of meropenem. At high inoculum these differences multiply and lead to significant differences as well as in MPC and MIC_HI_ values. Previously, the increased expression of levels of beta-lactamase genes in *K. pneumoniae* under the influence of meropenem was reported [37]. Similarly, as at MIC_HI_, assuming therapeutic plasma meropenem concentrations (after administration of a 1 g standard or 2 g high dose [34,35,36]), the success of meropenem therapy is highly probable against carbapenemase-negative meropenem susceptible *K. pneumoniae*. In contrast, if the key pathogen is a carbapenemase producer, there is a high probability of meropenem resistance due to the high MPC that exceeds peak meropenem concentration, and the antibiotic pharmacokinetic profile would be inside the MSW for most of the dosing interval.

As mentioned above, MPCs did not correlate with initial antibiotic MICs (the lower border of MSW) and, therefore, could not be predicted by the MIC (Figure 3). Similar conclusions about weak or absent correlations between the MIC and MPC have been reported with several classes of antibiotics [38,39]. Obviously, it would be very helpful at the initial stage of antimicrobial therapy, when the bacterial isolate is identified and screened for antibiotic susceptibility, if we could predict possible resistance-related treatment outcomes. In this light, we questioned whether, instead of standard MICs, the MIC determined at high bacterial inocula could be used for this purpose and if MIC_HI_ would correlate with the MPC, as both measures represent the “hidden” resistance potential. This might be important and practical because assessing MIC_HI_ (to characterize the “hidden” bacterial resistance potential) is much easier than MPC testing, which is not widely available in routine laboratory testing.

The results showed that MPC correlated well with MIC_HI_ (in contrast to MIC): the higher the MIC_HI_ the higher the MPC (Figure 4). The combined data for all *K. pneumoniae* strains were fitted by a single sigmoid function with a very high correlation coefficient (*r*^2^ = 0.93). This indicates that the MIC_HI_ can be used to predict the isolate’s MPC (independent of the production of carbapenemases). Furthermore, along with MPC, MIC_HI_-based pharmacokinetic/pharmacodynamic (PK/PD) indices, such as the percent of time when antibiotic concentration exceeds the MIC within the dosing interval, %T > MIC_HI_ [40] and the ratio of the area under the concentration versus time curve (AUC) to the MIC_HI_, AUC/MIC_HI_, can be used to construct “resistance-concentration” relationships. Using such relationships, it is possible to determine antibiotic regimens that could suppress resistance development. Therefore, we propose a series of in vitro studies using dynamic models to delineate relationships between MIC_HI_-based PK/PD indices and the development of meropenem resistance in carbapenemase-producing and non-producing *K. pneumoniae*.

In summary, the high MIC_HI_ and MPC values for carbapenemase-producing *K. pneumoniae* predict low meropenem efficacy against high-burden infections caused by these pathogens.

The study has several limitations: it did not include bacterial strains other than *K. pneumoniae* or other beta-lactam antibiotics that could generalize the current conclusions.

## 4. Materials and Methods

### 4.1. Antimicrobial Agent and Bacterial Strains

Meropenem powder was purchased from Sigma-Aldrich (St. Louis, MO, USA). Thirty-nine in total *K. pneumoniae*, including clinical isolates and ATCC collection strains (700603 and KPC reference strains: BAA 1705 and BAA 1898, BAA 1899, BAA 1900, BAA 1902, BAA 1904, and BAA 1905), with different susceptibility to meropenem in range from susceptible to resistant (according to EUCAST recommendations to use meropenem susceptibility MIC breakpoint of ≤8 mg/L [34]) were used in the study. Ten of them were carbapenemase non-producing, ten by each of *bla*_OXA-48_ and *bla*_NDM_ positive by PCR and nine of *bla*_KPC_ positive by PCR. Before each testing, carbapenemase production was double-checked for each bacterial strain by a modified carbapenem-inactivation method [41]. Thirty-one clinical isolates were collected from ICU patients admitted to the Moscow and Saint Petersburg hospitals. The sites of isolation were blood, sputum, urine and tracheal aspirate.

### 4.2. Susceptibility Testing

Susceptibility testing of *K. pneumoniae* to meropenem at standard inocula (SI) of approximately 5 × 10^5^ CFU/mL was performed using broth microdilution technique with a series of two-fold dilutions according to the CLSI recommendations [42]. A similar technique was used for susceptibility testing at high inocula of 5 × 10^7^ CFU/mL (HI). For all MIC testing experiments, the Mueller–Hinton broth (MHB) (Becton Dickinson, San Jose, CA, USA) was used. When the meropenem MICs were determined at HI, bacterial growth was quantified by optical density at 600 nm (OD), ODs before and after 18 h incubation at 37 °C were estimated. The MIC was the dilution at which the 18 h OD was equal to or less than that at time 0. An inoculum effect was defined as an eightfold or greater increase in MIC when tested with the HI and compared to that for SI [13]. MIC testing was repeated at least three times each with double replicates, and then the modal MICs were estimated.

The MIC breakpoint for meropenem susceptibility was used according to EUCAST recommendations [34]. The interpretive criteria for susceptibility were as follows: susceptible, ≤2 mg/L; resistant, >8 mg/L.

### 4.3. Mutant Prevention Concentration (MPC) Determinations

The meropenem MPC for each *K. pneumoniae* isolate was determined as described elsewhere [43,44]. Briefly, the tested microorganism was cultured in MHB and incubated for 24 h. Then, the bacterial suspension was centrifuged (4000× *g* for 20 min) and, after the removing of the supernatant, the pellet was re-suspended in MHB to yield a concentration of ~10^10^ CFU/mL. A series of agar plates (Mueller–Hinton agar) containing meropenem concentrations ranging from 0.03 to 8192 mg/L were then inoculated with a ~10^10^ CFU/mL suspension of *K. pneumoniae*, incubated for 24–48 h at 37 °C and screened visually for growth. The plates with a weak or doubtful bacterial growth were additionally stroked to the agar plates with similar meropenem concentration and incubated for 24–48 h. Then, the presence or absence of microbial growth was estimated. The MPC was taken as the lowest meropenem concentration that completely inhibited growth. The lower limit of detection was 10 CFU/mL (equivalent to at least one colony per plate). The MPC determination was repeated for each *K. pneumoniae* strain at least in triplicate.

### 4.4. Statistical Analysis

The reported MIC, MIC_HI_ and MPC data were obtained by calculation of the respective modal values.

The relationship between the MIC and MIC_HI_ data (both expressed in decimal logarithms) was fitted by the sigmoid function:*Y* = *Y*_0_ + *a*/{1 + exp[−(*x* − *x*_0_)/*b*]}(1)
where *Y* is log MIC_HI_; *Y*_0_ is minimal value of *Y*; *x* is log MIC; *a* is maximal value of *Y*; *x*_0_ is *x* corresponding to *a*/2; *b* is a parameter reflecting sigmoidicity.

The relationships between the MIC and MIC_HI_/MIC ratio and between the MIC_HI_ and MPC (all expressed in decimal logarithms) were fitted by the sigmoid function:*Y* = *a*/{1 + exp[−(*x* − *x*_0_)/*b*]}(2)
where *Y* is log MIC_HI_/MIC ratio or log MPC; *x* is log MIC or log MIC_HI_; *a* is maximal value of *Y*; *x*_0_ is *x* corresponding to *a*/2; *b* is a parameter reflecting sigmoidicity.

All calculations were performed using SigmaPlot 12 software (Systat Software Inc., headquartered in San Jose, CA, USA).

## 5. Conclusions

The current study shows low IE probability with carbapenemase-non-producing *K. pneumoniae* and high IE probability with carbapenemase-producing strains. The IE probability can be assumed from the MIC of a given *K. pneumoniae* strain using established “MIC_HI_/MIC ratio—MIC” relationships. We determined the general trend—the higher the meropenem MIC, the less meropenem activity will diminish at high bacterial burdens. Among tested carbapenemase-producing strains, OXA-48 producers exhibited the lowest inoculum-related MIC shifts, KPC- and NDM-producing bacteria exhibited quite high meropenem MICs at high inocula and MIC did not correlate with the MPC values. In contrast, a significant correlation was observed between the MIC_HI_s and the MPC, indicating that these bacteria-antibiotic characteristics are inter-related and describes the “hidden” bacterial resistance potential. To determine the possible resistance-related risk of failure outcome due to a given *K. pneumoniae* strain, we propose determining the MIC_HI_. This can more or less predict the MPC value of the particular strain. Along with MPC, MIC_HI_ (as part of PK/PD indices) can be used to construct “resistance-concentration” relationships.

## Figures and Tables

**Figure 1 antibiotics-12-00872-f001:**
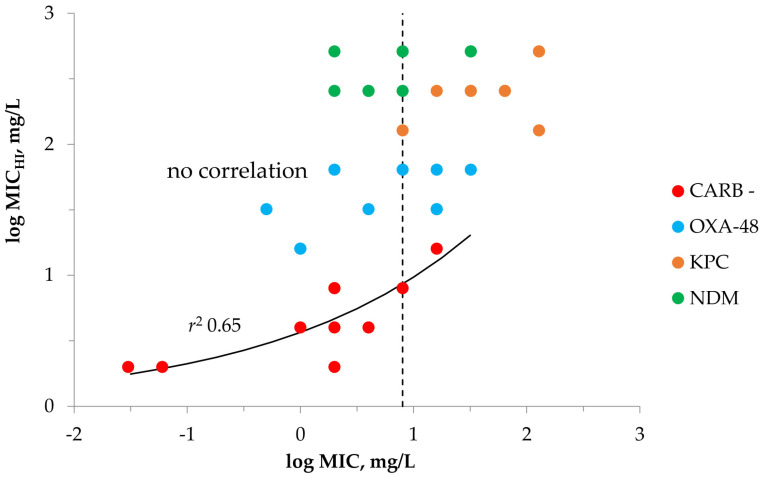
MIC_HI_ versus MIC data with meropenem and *K. pneumoniae.* Vertical dashed line indicates meropenem susceptibility MIC breakpoint of 8 mg/L. Data for carbapenemase-non-producing strains were fitted by Equation (1): *Y*_0_ = 0.008, *x*_0_ = 20.4526, a = 57332, b = 1.7718. “CARB−“is carbapenemase-negative group of strains; “OXA-48” is OXA-48-producing group of strains; “KPC” is KPC-producing group of strains; “NDM” is NDM-producing group of strains.

**Figure 2 antibiotics-12-00872-f002:**
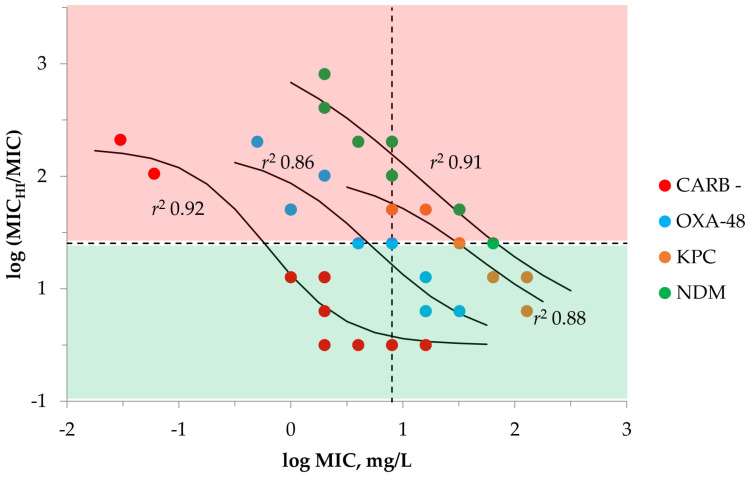
“MIC_HI_/MIC ratio—MIC” relationships with meropenem and *K. pneumoniae* fitted by Equation (2): *x*_0_ = −0.2134, *a* = 1.7495, *b* = −0.3573 (CARB-); *x*_0_ = 0.7329, *a* = 1.7362, *b* = −0.4657 (OXA-48 producers); *x*_0_ = 1.6671, *a* = 1.5557, *b* = −0.5255 (KPC producers); *x*_0_ = 1.2069, *a* = 2.86, *b* = −0.8116 (NDM producers). Vertical dashed line indicates the meropenem susceptibility MIC breakpoint of 8 mg/L. Horizontal dashed line indicates the threshold MIC_HI_/MIC ratio equal to 8 and associated with the IE. Red and green areas indicate the MIC_HI_/MIC levels are associated and not associated with IE, respectively. “CARB−“is carbapenemase-negative group of strains; “OXA-48” is OXA-48-producing group of strains; “KPC” is KPC-producing group of strains; “NDM” is NDM-producing group of strains.

**Figure 3 antibiotics-12-00872-f003:**
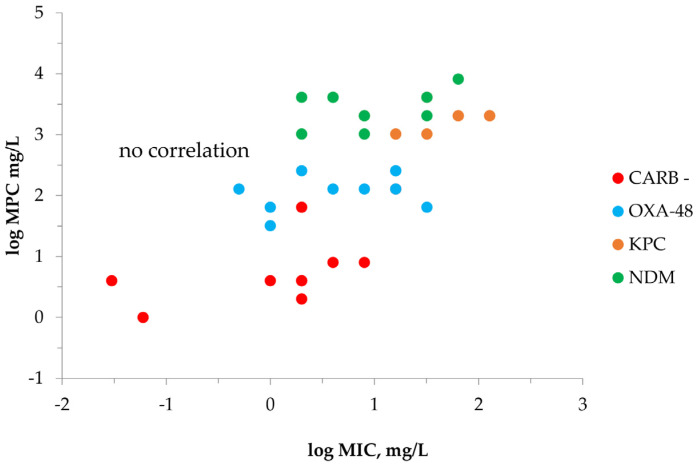
MPC versus MIC data with meropenem and *K. pneumoniae.* “CARB−“is carbapenemase-negative group of strains; “OXA-48” is OXA-48-producing group of strains; “KPC” is KPC-producing group of strains; “NDM” is NDM-producing group of strains.

**Figure 4 antibiotics-12-00872-f004:**
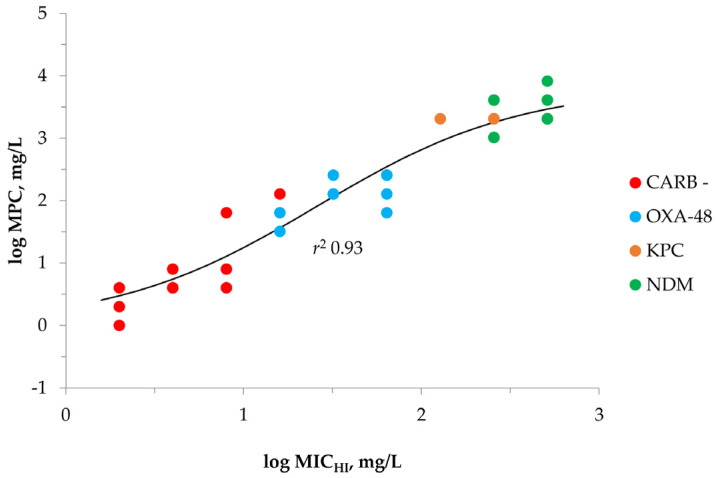
The “MPC—MIC_HI_” relationship with meropenem and *K. pneumoniae* fitted by Equation (2): *x*_0_ = 1.4167, *a* = 3.8256, *b* = 0.5702. “CARB−“is carbapenemase-negative group of strains; “OXA-48” is OXA-48producing group of strains; “KPC” is KPC-producing group of strains; “NDM” is NDM-producing group of strains.

**Table 1 antibiotics-12-00872-t001:** MICs (mg/L) at standard (MIC) and high inocula (MIC_HI_) data, MIC_HI_/MIC ratios and MPCs (mg/L) of meropenem against carbapenemase-producing and carbapenemase-non-producing *K. pneumoniae* strains.

Carbapenemase	*K. pneumoniae*	MIC	MIC_HI_	MIC_HI_/MIC	IE ^1^	MPC
Carbapenemase-negative (*n* = 10)	700,603	0.03	2	64	+	4
	188	0.06	2	32	+	1
	782	1	4	4	−	4
	2286	2	2	1	−	2
	2684	2	4	2	−	4
	3093	2	8	4	−	64
	3101	2	8	4	−	4
	1676	4	4	1	−	8
	844	8	8	1	−	8
	2895	16	16	1	−	128
OXA-48 carbapenemase producers (*n* = 10)	1278	0.5	32	64	+	128
	1128	1	16	16	+	32
	215	1	16	16	+	64
	1456	2	64	32	+	256
	1170	4	32	8	+	128
	3111	8	64	8	+	128
	202	16	32	2	−	128
	38	16	32	2	−	256
	485	16	64	4	−	256
	75	32	64	2	−	64
KPC carbapenemase producers (*n* = 9)	BAA 1904	8	128	16	+	2048
	BAA 1705	16	256	16	+	1024
	BAA 1900	32	256	8	+	1024
	14	32	256	8	+	4096
	BAA 1898	64	256	4	−	2048
	BAA 1899	64	256	4	−	2048
	BAA 1902	128	256	2	−	2048
	BAA 1905	128	512	4	−	2048
	16	128	512	4	−	2048
NDM carbapenemase producers (*n* = 10)	1326	2	256	128	+	1024
	2228	2	512	256	+	4096
	2342	4	256	64	+	4096
	35	4	256	64	+	4096
	1167	8	256	32	+	1024
	2131	8	512	64	+	2048
	3204	8	512	64	+	2048
	1961	32	512	16	+	4096
	3166	32	512	16	+	2048
	2863	64	512	8	+	8192

^1^ IE—inoculum effect, ≥8-fold MIC increase at high inocula of 5 × 10^7^ CFU/mL (HI); “+” indicates the IE was observed; “−“ indicates the IE was not observed.

## Data Availability

Not applicable.

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
