# Peer review of "Meropenem MICs at Standard and High Inocula and Mutant Prevention Concentration Inter-Relations: Comparative Study with Non-Carbapenemase-Producing and OXA-48-, KPC- and NDM-Producing Klebsiella pneumoniae"

_antibiotics, 2023, doi:10.3390/antibiotics12050872_

Round 1
Reviewer 1 Report
1. Line number 124. As the results shown in Figure 1, the interpretation is supposed to be "non-carbapenemase producing isolates did not show correlations between meropenem MICs at the two inoculum densities."
2. In Table 1, the MPC ranges among carbapenemase-producing K. pneumoniae vary depending on the different types (for example, 32–256 mg/L for OXA-48, 1024–4096 mg/L for KPC, and 1024–8129 mg/L for NDM). Could you provide more discussion about the relationship between MPC levels and specific carbapenemase producers?
3. The reason of using a population size of 10 10 CFU/ml cells to determine the lowest meropenem concentration for growth inhibition should be stated in this study (or provide references of the cell concentration that was used for the MPC determination).
Author Response
Response to Reviewer 1 Comments
Comments and Suggestions for Authors
Point 1. Line number 124. As the results shown in Figure 1, the interpretation is supposed to be "non-carbapenemase producing isolates did not show correlations between meropenem MICs at the two inoculum densities."
Response 1. We agree with the Reviewer’s comment that the “MICHI-MIC” correlation was too low to consider it meaningful. We refined the sentence at lines 125-128 of the revised manuscript: “The poor sigmoid relationship between the MIC and MICHI was observed only with carbapenemase non-producing K. pneumoniae with r2 = 0.65. However, the low correlation coefficient indicates that non-carbapenemase producing isolates did not show significant correlations between meropenem MICs at either inoculum densities.”
Point 2. In Table 1, the MPC ranges among carbapenemase-producing K. pneumoniae vary depending on the different types (for example, 32–256 mg/L for OXA-48, 1024–4096 mg/L for KPC, and 1024–8129 mg/L for NDM). Could you provide more discussion about the relationship between MPC levels and specific carbapenemase producers?
Response 2. We provided a possible explanation of relatively high MPC values for KPC and NDM producing isolates compared to the OXA-48 producers in the discussion section (lines 249-256 in the revised manuscript).
Point 3. The reason of using a population size of 1010 CFU/ml cells to determine the lowest meropenem concentration for growth inhibition should be stated in this study (or provide references of the cell concentration that was used for the MPC determination).
Response 3. We used a traditional MPC determination technique with an inoculum density of 1010 CFU/ml. The respective references were cited (ref 40 and 41) in the Materials and methods section (lines 312-313 in the submitted manuscript). In fact, high inoculum density is used to detect all possible antibiotic-resistant cells that can arise in the total bacterial population to account for possible emergence of resistance during antibacterial treatment of high-density infections.
Reviewer 2 Report
Dear Authors,
I've found your paper of merit. AMR is a critical issue and help clinicians to better interpret resistance tests by providing further evidence is so much insightful.
I would just improve introduction and discussion, including some data from the epidemiological and clinical point of view.
Please consider:
1. https://pubmed.ncbi.nlm.nih.gov/28603227/
2. 2022 Apr;28(4):425-435. doi: 10.1089/mdr.2021.0109. Epub 2021 Dec 15.
None
Author Response
Comments and Suggestions for Authors
I've found your paper of merit. AMR is a critical issue and help clinicians to better interpret resistance tests by providing further evidence is so much insightful.
Point 1. I would just improve introduction and discussion, including some data from the epidemiological and clinical point of view.
Please consider:
- https://pubmed.ncbi.nlm.nih.gov/28603227/
- 2022 Apr;28(4):425-435.doi: 10.1089/mdr.2021.0109. Epub 2021 Dec 15.
Response 1. We thank the Reviewer for the positive feedback and the reasonable suggestion to reflect antibiotic-resistant K. pneumoniae epidemiology; we considered the respective studies in the introduction section (lines 79-80).
Reviewer 3 Report
This is an excellent study that provides the significancy different measures of resistance profile of bacteria and correlation between and among them. It significantly provide an in-depth analysis of these parameters in relation with meropenem susceptibility, production of carbapenemases and specific types, and proposes determining the MICHI., which reasonably predict MPC value
Author Response
Comments and Suggestions for Authors
Point 1. This is an excellent study that provides the significancy different measures of resistance profile of bacteria and correlation between and among them. It significantly provide an in-depth analysis of these parameters in relation with meropenem susceptibility, production of carbapenemases and specific types, and proposes determining the MICHI., which reasonably predict MPC value.
Response 1. We are grateful to the Reviewer’s positive comment and hope our study will contribute to research in antibiotic resistance.
Reviewer 4 Report
In this manuscript, authors analyze the resistance profiles of K. pneumoniae strains that vary in their susceptibility to meropenem, carbapenemase production, and specific carbapenemase types. The authors also analyzed the interrelations between MIC, MICHI, and MPC for each tested strain. The study found low IE probability was observed in carbapenemase-non-producing K. pneumoniae strains, while high IE probability was detected in carbapenemase-producing strains. The authors also observed that MICs did not correlate with MPCs, but a significant correlation was observed between the MICHIs and MPCs, indicating that these bacteria/antibiotic characteristics display similar resistance properties of a given bacterial strain. Overall, the study provides valuable insights into the different measures of bacterial resistance and their role in predicting the clinical efficacy of antibiotics. The study's findings suggest that determining the MICHI can be a useful predictor of MPC values for a particular K. pneumoniae strain, which can help determine the possible resistance-related risk. However, it is essential to note that the study's findings are specific to K. pneumoniae strains and may not be generalizable to other bacterial species. The manuscript is well-written, and the experiments are well-designed and well-executed. I recommend the publication of this manuscript in its original form.
Author Response
Comments and Suggestions for Authors
Point 1. In this manuscript, authors analyze the resistance profiles of K. pneumoniae strains that vary in their susceptibility to meropenem, carbapenemase production, and specific carbapenemase types. The authors also analyzed the interrelations between MIC, MICHI, and MPC for each tested strain. The study found low IE probability was observed in carbapenemase-non-producing K. pneumoniae strains, while high IE probability was detected in carbapenemase-producing strains. The authors also observed that MICs did not correlate with MPCs, but a significant correlation was observed between the MICHIs and MPCs, indicating that these bacteria/antibiotic characteristics display similar resistance properties of a given bacterial strain. Overall, the study provides valuable insights into the different measures of bacterial resistance and their role in predicting the clinical efficacy of antibiotics. The study's findings suggest that determining the MICHI can be a useful predictor of MPC values for a particular K. pneumoniae strain, which can help determine the possible resistance-related risk. However, it is essential to note that the study's findings are specific to K. pneumoniae strains and may not be generalizable to other bacterial species. The manuscript is well-written, and the experiments are well-designed and well-executed. I recommend the publication of this manuscript in its original form.
Response 1. We are grateful to the Reviewer’s positive feedback and hope our study will contribute to ongoing research in antibiotic resistance. We agree with the Reviewer’s comment that the study results can be applied only to K. pneumoniae species and that similar studies with other beta-lactamase producing gram-negative and gram-positive bacteria are needed to explore if these results can be generalized to other bacterial species and antibiotics.